# Residential Radon Exposure in Patients with Advanced Lung Cancer in Lublin Region, Poland

**DOI:** 10.3390/ijerph19074257

**Published:** 2022-04-02

**Authors:** Anna Grzywa-Celińska, Izabela Chmielewska, Adam Krusiński, Krzysztof Kozak, Jadwiga Mazur, Dominik Grządziel, Katarzyna Dos Santos Szewczyk, Janusz Milanowski

**Affiliations:** 1Chair and Department of Pneumonology, Oncology and Allergology, Medical University of Lublin, 20-090 Lublin, Poland; izabelachmielewska@gmail.com (I.C.); adak93@gmail.com (A.K.); janusz.milanowski@umlub.pl (J.M.); 2Laboratory of Radiometric Expertise, Institute of Nuclear Physics PAN, 31-342 Cracow, Poland; krzysztof.kozak@ifj.edu.pl (K.K.); jadwiga.mazur@ifj.edu.pl (J.M.); dominik.grzadziel@ifj.edu.pl (D.G.); 3Department of Pharmaceutical Botany, Medical University of Lublin, 20-093 Lublin, Poland; k.szewczyk@umlub.pl

**Keywords:** residential radon, lung cancer, exposure

## Abstract

Exposure to radon is the second most common factor causing lung cancer in smokers and the first among non-smokers. We aimed to measure the impact of the radon exposure on patients with different histological types of advanced lung cancer. The measurement of radon exposure was performed in 102 patients with lung cancer in stage 3B or higher (Poland). There were 78.4% of patients with non-small cell carcinoma and 21.6% of patients with small cell carcinoma. One month radon exposure measurement was performed with trace detectors in order to control whether high radon concentrations (>800 Bq/m^3^) were found in the homes of patients with cancer diagnosed. Results of the determinations were then compared with the representation of the most common types of lung cancer in the study population. In the analyzed group, the average concentration of radon during the exposure of the detector in the residential premises of the respondents accounted for 69.0 Bq/m^3^ [37.0–117.0] and had no statistically significant effect on the type of lung cancer developed in patients. The lack of statistical significance may result from the small study group and the accompanying exposure to other harmful components. As the incidence of lung adenocarcinoma is increasing and exposure to tobacco smoke is decreasing, the search for other modifiable causes of lung cancer should be the task in the future.

## 1. Introduction

Lung cancer is the most common malignancy in terms of incidence and death from all oncological causes [1]. In 2020, an estimated 2.2 million new cancer cases were diagnosed causing 1.8 million deaths [2]. Non-small cell lung cancer (NSCLC) constitutes about 85% of all lung cancer. Its main subtypes include adenocarcinoma, squamous cell carcinoma, and large cell carcinoma [3]. Although it may develop on a genetic basis, it is mainly associated with exposure to harmful inhaled substances. The main substance is tobacco smoke and radon secondly. About 1 in 7 of the deaths from radon-related lung cancer is caused by radon and not by active smoking. Therefore, radon remains the main cause of cancer in non-smokers [4]. However, there is an undeniable synergistic effect between these two factors [5]. Since there is an increasing proportion of non-smokers in NSCLC cases, there is an urgent need to investigate it from a wider perspective [6,7]. Other risk factors include air pollution, exposure to dust, especially silica and asbestos, infectious agents, and chronic and inflammatory lung diseases such as idiopathic pulmonary fibrosis and tuberculosis. Taking this into consideration, many cases of lung cancer could be prevented.

A review of the literature shows the differentiation of histological types of lung cancer depending on the environmental exposure factor. The link between cigarette smoking and squamous cell carcinoma and small cell carcinoma is best known. In the case of air pollution, i.e., smog, the observations showed a relationship between exposure to ozone and squamous cell carcinoma in particular, while the exposition to PM_2.5_ caused small-cell and adenocarcinoma, and nitrogen dioxide (NO_2_), especially adenocarcinoma [8]. Among the lung diseases predisposed to the development of lung cancer, a stronger association between tuberculosis with glandular type [9], emphysema, chronic bronchitis and asthma with squamous cell [10], and idiopathic pulmonary fibrosis (IPF) with both glandular and squamous cells [11] was noted. In the case of radon, studies have reported its effect on all histopathological types of lung cancer, in particular adenocarcinoma [12].

Radon (Rn-222) is a naturally occurring radioactive element from the group of the noble gases. It belongs to the uranium (U-238) path, its half-life amounts to 3.82 days, and when decaying into other radioactive derivatives, it emits mainly α (alpha) radiation. This radiation has a low range in tissues, but a high potential to damage DNA—when radon enters the bronchial tree, alpha radiation can initiate carcinogenesis. The main source of radon in homes and workplaces is the ground on which the building stands. Building materials, water and natural gas are other sources of indoor radon. In Poland, this gas reaches high concentrations mainly in the Sudetes, where the substrate is rich in radioactive uranium [13]. However, elevated indoor radon levels may be a local problem, due to the permeability of soil or the presence of local disjunctive tectonic zones. The quality of building construction and the tightness of installations (electrical and gaseous) also influence indoor radon levels. Radon concentration in the air of dwellings is subject to seasonal and twenty-four hour variations.

The reference level of radon concentration in buildings, as well as permissible radon concentration in water are regulated by law. Apart from that, there is a regulation concerning the content of radium (Ra-226) in building materials, which is a direct predecessor of gaseous radon Rn-222. Additionally, there are individual possibilities of reducing the exposure to this element, mainly through an appropriate ventilation of houses [14,15,16]. Radon is therefore another environmental risk factor for lung cancer, and its concentration in the vicinity of humans can be modified by appropriate behavior or lifestyle.

The carcinogenic effect of radon results from its radioactive properties. The relationship with lung cancer depends on its gaseous state of aggregation, since radon reaches deep into the lower respiratory tract. It is deposited mainly in the area of bronchial bifurcations and decomposes into further radioactive derivatives, of which it is a particularly dangerous isotope of Po-218 polonium, emitting α (alpha) and β (beta) particles. The most harmful type of radiation from radon and its derivatives are alpha particles, which have high energy but can be easily shielded. However, if they reach the lungs, they have the ability to act directly on epithelial cells, where they exert a strong carcinogenic effect. Ionizing radiation damages the genetic material directly, by breaking the double-stranded structure of DNA, but also indirectly, as the result of water radiolysis, which generates reactive oxygen species. Research has shown the compound of radon, among others with p53 mutations, increased expression of the KRAS oncogene and down-regulation let-7 microRNA. A slightly increased frequency of ALK gene rearrangements has also been reported in patients exposed to radon [8,16,17,18,19].

The aim of our study was to measure the exposure to residential radon in patients with lung cancer in the region of Lublin, Poland. The goal of these measurements was also to check whether the exposure to radon in the patients’ places of residence did not significantly exceed the reference level. We also tried to estimate the impact of exposure on the representation of major histological types of lung cancer to identify and characterize the different risk factors for lung cancer with special emphasis on residential radon.

If a high radon exposure was detected, an additional aim could be set, which could become the protection of the co-habitants remaining in these apartments, by proposing methods of reduction of radon exposure.

## 2. Materials and Methods

### 2.1. Study Group

The measurement of radon exposure was performed in 102 patients with patients with inoperable lung cancer stage 3B or higher—all patients were residents of the Lublin region (Figure 1). They were all treated at the Department of Pneumonology, Oncology and Allergology of the Medical University of Lublin, Poland. 

The study group consisted of 39 women (38.2%) and 63 men (61.8%). The majority of the analyzed group were patients over 65 years of age (*n* = 63; 61.8%), younger patients accounted for 38.2% of the study group. The division into individual histopathological types was also performed—the study group reflects the distribution of histological types of lung cancer in the Polish population [20].

Patients with the diagnosis of small cell carcinoma accounted for 21.6% (*n* = 22), and non-small cell carcinoma was diagnosed at 78.4% (*n* = 80) of patients. In the non-small cell carcinoma group, adenocarcinoma subtype occurred in 41.2% (*n* = 42), squamous subtype in 26.5% (*n* = 27), and not otherwise specified (NOS) due to an uncertain histological subtype in 6.9% (*n* = 7). Four patients were diagnosed with rare types lung cancer, described in the table as other. In this subgroup of patients, there were neuroendocrine, mixed histology of adenosquamous, and two patients with large cell tumors. Characteristics of the study group are presented in Table 1.

Most of the patients were smokers or ex-smokers, and in addition, many patients reported exposure to passive smoking and other harmful substances. The distribution of patient exposures is presented in Table 2.

### 2.2. Radon Exposure Measurements

Indoor radon concentration measurements were performed using a passive method with solid-state nuclear track detectors (SSNTD), type CR-39 (Radosys, Ltd., H-1116 Budapest, Hungary). In this study the detectors made by the Hungarian company—RadoSys (RSKS type Radon Dosimeter) have been applied. The detector consists of a plastic cover (diffusion chamber), inside which the CR-39 foil (PADC/CR-39 Plastics) is placed, which is the actual radon measuring element. The detector is placed in a sealed foil (detector packaging). The inside of the package is filled with pure nitrogen—this ensures that the detector is stored before exposure in a radon-free atmosphere. The detector exposure begins at the moment of breaking the foil and removing the detector from the packaging. The detectors were sent together with detailed instructions on how to expose them. The instruction contained the information on where the detector should be placed, i.e., in which room (mostly bedrooms and living room), in what distance from walls, ceiling (min. 20 cm), at what height (should be the height at which people mostly breathe, i.e., 120–180 cm above the floor), away from windows, doors, heaters. The instruction also contained a request not to move the detector into another place and not to lose it. Together with the detectors, after the exposure, people also sent back a short questionnaire about the conditions of the detector’s exposure. Measurements of radon concentration in residential buildings consisted of a 30-day exposure of the detector in a given room, and then, after removing the CR-39 foil from the diffusion chamber, the readings were made in the Laboratory of Radiometric Expertise, Institute of Nuclear Physics of the Polish Academy of Sciences in Cracow, Poland. Alpha particles, emitted by radon penetrating inside the detector through the aperture in the diffusion chamber, cause microscopic damage in the film, visible after etching. Etching of the CR-39 film is carried out in a 25% solution of sodium hydroxide (NaOH) in a special “chemical bath” (RadoBath, Radosys, Ltd., H-1116 Budapest, Hungary) under strictly controlled conditions: the etching temperature is 90 °C, and the etching time is 4 h. The traces left by the alpha particles emitted by radon appear as dark spots on the CR-39 film after etching. Computer analysis of the microscopic image of the detector surface (CR-39 foil) allows us to calculate the number of these holes. Knowing the so-called calibration factor it is possible to determine the radon concentration at the detector exposure site based on the density of the traces (i.e., the number of traces per 1 cm^2^ of CR-39 foil). The calibration factor is determined by exposing the detector to several known radon concentrations. Calibration is performed in the so-called radon chamber. Counting the number of traces per unit area of CR-39 film is performed by the automatic reading system of CR-39 detectors—the professional reader Radometer 2000 (Radosys, Ltd., H-1116 Budapest, Hungary). The reader is equipped with a charged-coupled device (CCD) camera and allows saving images on the computer disk with 100× magnification. The scanning area of the CR-39 detector is 47 mm^2^. During the scan, the detector code is automatically read. The increase in the number of traces in the field of view of the CCD camera corresponds to a higher average concentration of radon in the place where the detector was exposed. The described method used by the Laboratory of Radiometric Expertise, Institute of Nuclear Physics PAN in Cracow is defined in the scope of Accreditation No AB 788 (Polish Centre for Accreditation) and meets the requirements of the PN-EN ISO/IEC 17025:2018-02 standard.

Results of the determinations were then compared with the representation of the most common types of lung cancer in the study population. The additional variables such as an epidermal growth factor receptor (EGFR) mutations, ROS-1 and ALK rearrangement, and ligand expression for the programmed cell death receptor on cancer cells (PD-L1) were also taken into consideration in NSCLC patients.

### 2.3. Statistical Analysis

The statistical analysis of the data collected in the spreadsheet was performed using the MedCalc (v.15.8) and Statistica (v.13) software. Categorized data are presented numerically and as percentages. Due to the different than normal distribution of a given continuous—the average radon concentration during the detector exposure in the apartment of the examined person—it was presented using the medians and the interquartile range. The distribution of the variable mean radon concentration during the exposure of the detector in the apartment of the examined person was assessed using the D`Agostino-Pearson test.

## 3. Results and Discussion

In the analyzed group, the average concentration of radon during the exposure of the detector in the residential premises of the respondents had no statistically significant effect on the type of lung cancer developed in individual patients (*p* = 0.3696). The results did not differ when comparing the two main types of lung cancer (non-small cell carcinoma vs. small cell carcinoma: *p* = 0.8547). Similarly, the presence of EGFR gene active mutations, ALK gene rearrangement, or ROS-1 rearrangement (*p* = 0.7499) did not vary among patients with different levels of radon exposure (the presence of these mutations is routinely assessed only in patients with a non-squamous type of non-small cell lung cancer).

Detailed data on the comparison of the average concentration of radon during the exposure of the detector in residential premises in patients with lung cancer depending on the selected clinical variables are presented in Table 3.

It is worth noting that residential radon is an important but modifiable factor contributing to the occurrence of lung cancer. While smoking is undeniably the strongest risk factor for lung cancer, it is radon, this underappreciated killer, that is the second most common cause and the first among non-smokers [21]. It should also be noted that exposure to tobacco smoke poses higher risk to induce lung cancer in the population exposed to radon [22]. Numerous studies show that there is no safe level of exposure to radon [23].

Up till now, there is no large database for indoor radon in Poland. Based on available publications, the average concentration of indoor radon in Poland amounts to 49 Bq/m^3^ [24], which is lower than our measurements. It has been previously described that radon concentration might differ in certain parts of Poland.

There were many studies conducted to estimate the incidence of lung cancer associated with radon exposure [24,25]. A Norwegian study shows that radon may be responsible for 12% of lung cancer cases (the average concentration of radon in Norwegian homes in this study was 88 Bq/m^3^) [26]. The study by Krewski proved that the risk of lung cancer increases by 11% as the radon concentration increases by 100 Bq/m^3^ [27], although Darby et al. calculate the increase of this risk at a level of 8.4% [25].

For tobacco smoking, the strongest association was observed for squamous cell carcinoma (SQC) and small cell carcinoma (SCLC), followed by large cell cancer (LGC) and adenocarcinoma (ADC) [27]. The study of Barros-Dios et al. proved that the less frequently represented types of lung cancer, e.g., large cell carcinoma or microcellular carcinoma, had a greater representation among patients with a higher exposure to radon. Metaanalysis of 28 studies including 13,748 cases of lung cancer and 23,112 controls showed that radon exposure correlated with every histological type of lung cancer, with the strongest correlation for SCLC, followed by adenocarcinoma [28].

Our study did not reveal a significant correlation between the exposure to residential radon and the major histological types of lung cancer. This may be due to too small a study group. There were was a slight difference between frequent types of lung cancer and others. However, due to only four patients with other types of lung cancer, the difference was not significant. In addition, the radon exposure determinations in our group indicated a relatively low exposure. Another possible reason could be the observation that radon exposure was generally similar at all measurement sites in the study area. There were 82.4% of patients who smoked cigarettes (currently or in the past) and 35.3% reported second-hand smoking. In addition, 55% of patients reported exposure to various types of harmful substances (chemical, physical, biological, smoke, pesticides), therefore, the exposure to at least two factors could overlap.

Besides, radon exposure assessment has many limitations. First, there are significant differences between different places of residence, both within the same region and between regions. In addition, radon exposure can vary in the different levels of the same building. Residential radon concentrations depend on many factors, such as the type of soil, building materials and type of ventilation [29]. In addition, it should be noted that people who are subject to assessment of radon exposure can spend part of the day outside their home, e.g., at their workplace, where their exposure can be at completely different levels.

## 4. Conclusions

Our study was the first to explore radon as a potential lung cancer risk factor in the eastern part of Poland. Although exposure to radon in the study population was relatively low and averaged 69.0 [37.0–117.0], it exceeded the median radon exposure in Poland. It is worth noting that the highest levels were observed in patients with adenocarcinoma and patients with EGFR mutations. No statistically significant correlation was found between the exposure to radon and the incidence of particular histological types of lung cancer. The lack of statistical significance may result from the small study group. Since most of our patients were smokers or ex-smokers, it may influence the results as well. Moreover, most of the study reported having contact with other harmful biological and chemical factors. However, due to the increasing frequency of adenocarcinoma in parallel with the decreasing rate of smoking throughout Europe, especially among women, it is compulsory to search for other avoidable risk factors of lung cancer, and radon is considered one of them. In-depth, long-term measurements and large group studies should be implemented in this field [30].

## Figures and Tables

**Figure 1 ijerph-19-04257-f001:**
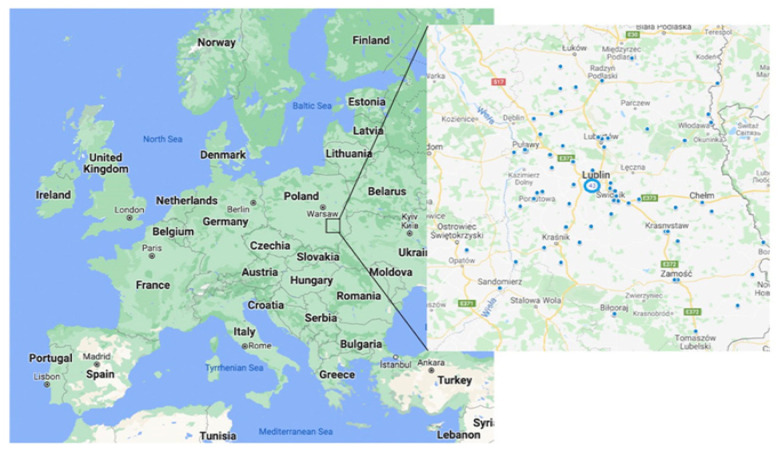
Distribution of radon exposure measurement sites in the Lublin region, Poland. Poland Google Maps. 2022. Lublin. 1:1,000,000. Google Maps [online]. Available online: https://www.google.com/maps (accessed on 1 February 2022).

**Table 1 ijerph-19-04257-t001:** Characteristics of the study group.

Variable		Study Group *n* = 102 (%)
Gender	Female	39 (38.2%)
	Male	63 (61.8%)
Age	<65	39 (38.2%)
	≥65	63 (61.8%)
Occupation	Engineer	7 (6.9%)
	Driver	6 (5.9%)
	Teacher	4 (3.9%)
	Worker	17 (16.7%)
	White-collar worker	19 (18.6%)
	Farmer	17 (16.7%)
	Craftsmen	12 (11.8%)
	Health professional	8 (7.8%)
	Salesman	5 (4.9%)
	Clerk	4 (43.9%)
	Other	3 (2.9%)

**Table 2 ijerph-19-04257-t002:** Smoking status and occupational exposure to harmful substances in patients of the examined group.

Factor		Study Group *n* = 102 (%)
Smoking status	current smoker	35 (34.3%)
	ex-smoker	49 (48%)
	non-smoker	18 (17.6%)
Second-hand smoking exposure	Yes	36 (35.3%)
	No	66 (64.7%)
Place of second-hand smoking exposure	Home	19 (54.3%)
	Workplace	10 (28.6%)
	Other	6 (17.1%)
Exposure to harmful substances	Yes	55 (53.9%)
	No	47 (46.1%)
Type of harmful substances	Biological	
	chemicals *	4 (7.4%)
	physical **	17 (30.7%)
	Dusts	8 (14.8%)
	Pesticides	10 (18.5%)
	no data (*n* = 48)	15 (27.8%)

* E.g., grease, glue, disinfectant, cleaning supplies; ** E.g., humidity, high or low temperature, welding.

**Table 3 ijerph-19-04257-t003:** The average radon concentration during detector exposure in a residential environment in lung cancer patients depending on selected clinical variables.

Variable		Average Radon Concentration during Detector Exposure [Bq/m^3^]	Study Group *n* = 102 (%)	*p*
Overall exposure to radon in the study group		69.0 [37.0–117.0]	*n* = 102	
Representation of major types of lung cancer	Non-small cell lung cancer (NSCLC)	72.5 [36.0–118.0]	80 (78.4%)	0.8547
	Small cell lung cancer (SCLC)	66.5 [45.0–86.0]	22 (21.6%)	
Representation of subtypes of NSCLC	Adenocarcinoma	70.5 [37.0–100.0]	42 (41.2%)	0.3696
	Squamous cell lung cancer	69.0 [32.0–119.0]	27 (26.5%)	
	Not-Otherwise-Specified (NOS)	58.0 [35.5–108.7]	7 (6.9%)	
	Other	108.5 [95.0–147.5]	4 (3.9%)	
Genetic aberrations EGFR/ALK/ROS-1	Yes	74.0 [34.2–119.2]	15 (34.9%)	0.7499
	No	66.0 [38.5–97.5]	28 (65.1%)	

EGFR/ALK/ROS-1: epidermal growth factor receptor/anaplastic lymphoma kinase/proto-oncogene tyrosine-protein kinase.

## Data Availability

Not applicable.

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
