# Peer review of "Residential Radon Exposure in Patients with Advanced Lung Cancer in Lublin Region, Poland"

_ijerph, 2022, doi:10.3390/ijerph19074257_

Round 1

Reviewer 1 Report

The subject of the submitted article - exposure to radon - is a significant contemporary problem, and from the substantive point of view, the work is well in the field of IJERPH, Environmental Health section, Special Issue “Environmental Radioactivity Monitoring and Measurements: Radon”.

Based on monthly measurements of radon concentration, the authors determined the mean value of radon exposure and tried to find a correction between different mean values of the radon concentration and the occurrence and severity of small cell lung cancer or non-small cell lung cancer, also taking into account different kinds of non-small cell lung cancer. The authors measured that the average value of the radon concentration in the area covered by the measurement (south-eastern part of Poland) was relatively low (about 69 Bq/m3), although it exceeded the median radon exposure in Poland.  Unfortunately, no statistically significant correlation was found between the exposure to radon and the incidence of individual histological types of lung cancer. This, let's call it a 'negative' result certainly made discussion difficult, nevertheless it is a result of research and it is worth for presentation. However, the article is not suitable for publication in its current version.

So, I suggest to accept the work for publishing after major corrections:

(1) Introduction (and also Discussion): the Introduction is incomplete, because most of the text in the Discussion section (lines 155-212) is also literature data informing about the current state of knowledge. I recommend to move this fragment to the Introduction section.

(2) Ad Table 3:

  1. Table title: “Comparison of the average radon concentration during detector exposure in a residential environment in lung cancer patients depending on selected clinical variables” - I would like to point out that it is not the radon concentration that depends on various clinical cases, but if there is a relationship, then clinical variables can depend on radon concentration;
  2. Based on the data in column “Averge radon concentration during detector exposure [Bq/m3], it can be said that radon exposure is generally similar at all measurement sites in the study area. And maybe this is also the reason why radon exposure (in this research) did not have a noticeable effect on the type and severity of lung cancer in the studied patients;
  3. When analyzing the data in this column, it can be seen that the only noticeable difference in radon exposure is for the subtype of NSCLC “other”. Perhaps it would be worth discussing this in more detail (what types of lung cancer are there) in the Discussion section;
  4. Please clarify the difference (which lung cancer subtypes belong to these two groups) between the terms “NOS” and “other”.

(3) The work also has a number of minor shortcomings, e.g.:

  • lines 73 and 78: it is exactly the same information;
  • line 104: “Institute of Nuclear Physics PAN in Kraków”;
  • line 107: solution of NaOH sodium;
  • lines 235-236: “Radon exposure and smoking have synergistic oncogenic effects” – this is a known issue and is not really a conclusion from the research done;
  • a abbreviation list would also be handy.

Reviewer 2 Report

Dear Athour, 

he manuscript is not in a good format and structure. Not such new data shows. from 2 pages discussion about the research finding just 5 sentence was belong to this research and the rest was from literature which are belong to introduction. not relevent and suitable method was used for this type of research. The manuscript is not well-written, I regret to say in my opinion it should be reject. False statments, fulfilling the manuscript with literature and repeating several times. 

find my comments in attached file.

Regards,

Reviewer 3 Report

Please, see the attached file

Round 2

Reviewer 1 Report

Substantially  the manuscript has been revised.

Reviewer 2 Report

it was improved.

Reviewer 3 Report

Dear authors,

my congratulations. After the review phase, the paper has been well improved.